# Antioxidant Content of Aronia Infused Beer

**Alexander Jahn [1],\*** , **Juyeong Kim [2]** , **Khawaja Muhammad Imran Bashir [1]** and **Man gi Cho [2]**

[1]   German Engineering Research and Development Center LSTME Busan, 1276 Jisa-Dong, Gangseo-Gu, Busan 46742, Korea; Imran.bashir@lstme.org

[2]   Department of Bio and Chemical Engineering, Dongseo University, Busan 47011, Korea; young7037@gmail.com (J.K.); mgcho@dongseo.ac.kr (M.g.C.)

\*   Correspondence: alexander.jahn@lstme.org

**Abstract:** Beer is a fermented beverage in which antioxidants can contribute to the oxidative stability and nutraceutical properties of the product. Aronia berries are antioxidant-rich fruits of distinct sour and astringent taste, limiting their culinary uses. Previously, fermentation has proven to be effective in the removal of astringent tastes from various fruit juices. In this study, a single malt pale ale was produced and infused with Aronia berries under various process conditions by adding the antioxidant-rich fruits at different stages of the beer brewing process. The polyphenol content, antioxidant potential and color were determined. There was a positive correlation between the Aronia amount added and antioxidant capacity. Higher concentrations of added Aronia also increased the polyphenol content and EBC color rating, while no change in the resulting pH was observed. An increase in the Aronia amount increased the attenuation, showing a positive effect on sugar utilization during fermentation. The addition of Aronia after the boil yielded the highest coloration and antioxidative capacity, while addition before the boil yielded a similar antioxidative capacity with a lower EBC rating. Taken together, the infusion of pale ale with Aronia berries can increase the EBC rating, polyphenol content and antioxidative capacity of the beer.

**Keywords:** beer; Aronia; fermentation; antioxidant

## 1. Introduction

Beer is a fermented, often alcoholic beverage rich in carbohydrates, amino acids, minerals, vitamins and phenolic compounds. The later mostly originate from malt (70–80%) and to a smaller extent from hops [1]. Beer brewed in the lager style with bottom-fermenting low-temperature strains (3.3 to 13.0 °C) of *Saccharomyces pastorianus* represents more than 90% of the beer produced worldwide [2]. Well-known brands such as Oettinger produce the pale lager style. On the other hand, ales are typically fermented at warmer temperatures (20 ± 4 °C) with top-fermenting yeast strains, leading to the formation of esters and other secondary flavor and aromatic compounds [3].

The world market of beer is still dominated by traditionally brewed and mass-produced beers, and some countries allow only the basic ingredients (water, malt, hops and yeast). However, recently, a significant increase in the market volume for craft beer can be observed. This developing market is partially driven by novelty and shows much shorter innovation cycles than the conventional beer market [4,5]. This has led to a diversification of the available beer types and an active search for interesting new beers and flavors [6]. Fruit infusion and fruity beers are one such trend, which have recently gained popularity. Beer with fruit ingredients may exhibit additional flavors but also might benefit form additional or increased bioactive compounds, which might, among other effects, lead to an increase in the oxidative stability of the beverage.

Fruits have been used as beer adjuncts for centuries, especially with Belgian lambic sourbeers [7]. Many tropical, subtropical and temperate fruits have not only nutritional value but also consumer

health benefits as well, which can be sometimes traced back to the antioxidant activity of phenolic compounds [8]. Polyphenols are natural antioxidants playing significant roles in the human body due to their capability to inhibit free radicals, which may cause cell damage [9].

Aronia berries, the fruits of the shrub *Aronia melanocarpa*, are gaining popularity due to their high antioxidant contents. Anthocyanin concentrations of up to 1480 mg per 100 g of fresh weight have been reported [10]. The direct consumption of Aronia berries is not common as its high tannin content leads to strong astringency via protein binding and precipitation on the mucus membrane in the mouth [11].

Changes in astringency have been observed in various fermentative processes and products. Common interactions are the metabolization of the astringent compounds and their precipitation during fermentation. We hypothesized that the infusion of wort with Aronia berries during the brewing process leads to the extraction of antioxidants, which will still be present after fermentation. The intention of this study was to develop a beer enriched with Aronia berries to increase its antioxidant capacity and determine if the polyphenol-based antioxidative capacity persisted through the fermentative stage of the beer's production. To this end, a top-fermented ale was enriched with Aronia berries by adding different amounts, over different times, at different stages of the beer production process. The resulting beers were analyzed for their antioxidant capacity and polyphenol concentration.

## 2. Materials and Methods

### 2.1. Materials

Gallic acid, Folin & Ciocalteu's Phenol Reagent (FCR) and 2,2-dipheny-1-picrylhydrazyl (DPPH) were purchased from Sigma Aldrich (St. Louis, MO, USA). Ascorbic acid (99.5%), sulfuric acid (95%), ethyl alcohol anhydrous (99.9%), ethyl alcohol (94.5%) and methyl alcohol (99.9%) were acquired from Samchun Pure Chemical Co. Ltd. (Pyeongtaek, Korea). Sodium carbonate anhydrous (99%) was purchased from Daejung Chemicals & Metals Co. Ltd. (Siheung, Korea). Phenol (98%) was acquired from Junsei Chemical Co. Ltd. (Tokyo, Japan). Raw Aronia berries (Sunchang, Korea) without stalks were purchased locally in Busan (Korea) and kept in the original packaging in a refrigerator at 5 °C until use. Vienna malt from Weyermann (Bamberg, Germany), Hallertauer Herkules hops from Hopsteiner (Mainburg, Germany) and *Saccharomyces cerevisiae* Safale S-04 dry ale yeast from Fermentis were used for the wort production.

### 2.2. Brewing and Aronia Berry Infusion

The brewing process is outlined in Figure 1. The mash was prepared from Vienna malt, which was crushed to the appropriate size and mixed with water at a ratio of 1:5.25, resulting in 4 kg of malt mixed with 21 L of water in a commercial brewing plant from Shandong Zunhuang Brewing Equipment (Jinan, China). Mash-in was started at 51 °C for 15 min, with saccharification rests at 67 °C for 30 min and at 70 °C for 15 min, and mash-out was finished at 75 °C for 5 min. The wort was filtrated and separated from the spent grains by an installed metal mesh filter and additional filtration through a fabric. The wort was divided into batches of 800 mL and individually boiled for one hour, five batches were boiled with different concentration of Aronia berries (DCA) for 1 h, and 0.2 g of hops was added 10 min before the end of the boil. The other five batches of wort were boiled with 10 g of Aronia berries, added at different times during the boiling process (DBT). Here, 0.2 g of hops was also added to the boil 10 min before the end. The bitter wort was cooled and inoculated with 0.5 g of yeast. Primary fermentation was done at 18 °C. One batch received 10 g of Aronia berries after the boil, while to another batch, 10 g of Aronia berries were added after cooling. Another two batches received 10 g of Aronia berries after 1 day and 3 days of fermentation, respectively. The standard was prepared in the same way without any Aronia berries added. An overview of the process conditions and sample coding is given in Table 1.

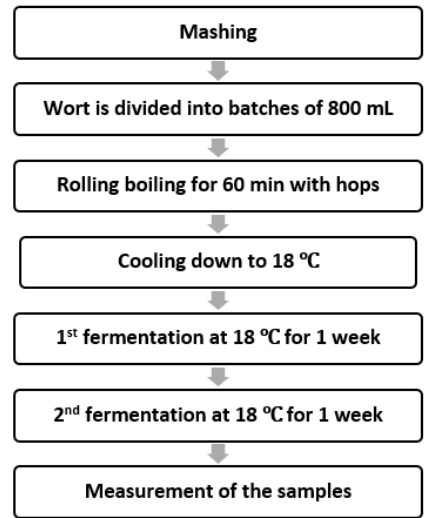

**Figure 1.** Flowchart of brewing process.

**Table 1.** Sample coding and infusion conditions.

| Abbreviation | Description | Aronia Concentration | Point of Addition |
|---|---|---|---|
| DCA 1.25 | Different Aronia Concentration | 1.25 g/L | Before the boil |
| DCA 6.25 | Different Aronia Concentration | 6.25 g/L | Before the boil |
| DCA 25 | Different Aronia Concentration | 25 g/L | Before the boil |
| DCA 62.5 | Different Aronia Concentration | 62.5 g/L | Before the boil |
| DCA 125 | Different Aronia Concentration | 125 g/L | Before the boil |
| DBT 60 | Different Boiling Time | 12.5 g/L | 60 min before end of the boil |
| DBT 45 | Different Boiling Time | 12.5 g/L | 45 min before end of the boil |
| DBT 30 | Different Boiling Time | 12.5 g/L | 30 min before end of the boil |
| DBT 15 | Different Boiling Time | 12.5 g/L | 15 min before end of the boil |
| DBT 5 | Different Boiling Time | 12.5 g/L | 5 min before end of the boil |
| BB | Before Boil | 12.5 g/L | Before the boil |
| AB | After Boil | 12.5 g/L | At the end of the boil |
| AC | After Cooling | 12.5 g/L | After wort was cooled to 18 °C |
| AF 1 | After Fermentation start | 12.5 g/L | 1 day after start of fermentation |
| AF 3 | After Fermentation start | 12.5 g/L | 3 days after start of fermentation |
| Standard | Standard | - | - |

## 2.3. Glucose Equivalent, pH and Color Analysis

The carbohydrate content of the beer was determined as the glucose equivalent with the phenol–sulfuric acid assay according Chow [12]. Beer samples were diluted 1 to 50 with water, and 50 μL of diluted sample was then placed in a 96 well plate and mixed with 150 μL of sulfuric acid and 30 μL of 5% phenol. After flotation in a heating bath (Heating Bath B-490, BÜCHI, Switzerland) for 5 min at 90 °C, the plate was cooled to room temperature and subsequently measured on a BioTek Epoch Microplate Spectrophotometer at λ = 490 nm (BioTek, Winooski, VT, USA) in triplicate. Glucose was used for the construction of a calibration curve. The color was measured with a UV-vis spectrophotometer at 430 nm according to the Standard Reference Method (SRM) in triplicate. The pH was measured from a unified sample with a pH meter (CyberScan pH 510, Eutech Instruments Pte Ltd., Singapore).

## 2.4. Total Polyphenol Content (TPC) Assay

The concentration of the total polyphenols in the beer samples was determined by the Folin–Ciocalteu colorimetric method. Folin and Ciocalteu's Phenol Reagent (FCR) was diluted with water 1:15 (v/v), 15 μL of sample was placed in a 96 well microplate, and diluted FCR was added. The mixture was incubated in the dark for 10 min at 18 °C. Subsequently, 15 μL of 20% sodium

carbonate was added to each well, the plate was agitated, and the absorbance was measured with a UV-vis spectrophotometer at λ = 755 nm. The results were compared with the gallic acid equivalent (GAE) in the concentration range of 25–300 μg/mL with the help of a calibration curve. The method was adopted without modification from Ducruet [1].

### 2.5. Ascorbic Acid Equivalent Antioxidant Capacity (AEAC) According to DPPH

The DPPH radical scavenging assay was adopted and modified from Sharma and Bhat [13]. Briefly, 10 mg of DPPH was dissolved in 250 mL of 80% methanol. Then, 20 μL of sample was placed in a microplate well and 300 μL of DPPH solution was added. The mixture was incubated in the dark for 30 min at 18 °C. The absorbance was measured with a UV-vis spectrophotometer at λ = 517 nm. The results were compared with a calibration curve from ascorbic acid in the range of 5–100 μg/mL.

### 2.6. Statistical Analysis

One-way analysis of variance for single factors was performed for relevant data. Unless otherwise noted, a $p < 0.05$ was considered statistically significant. Correlation analysis was performed with Pearson correlation coefficients, and all analyses were performed in Microsoft Excel 16.0.

## 3. Results and Discussion

### 3.1. Carbohydrate Content

Non-fermented sugars in the finished beer contribute to the nutritional value of the beer and to its sweetness. Here, the phenol–sulfuric acid assay was used for the determination of the remaining carbohydrates as glucose equivalents in the finished beer. While complex and various sugars have different response values for this assay [14], it is nonetheless a good indication of the remaining sweetness, ethanol production and completeness of fermentation or "attenuation". Decreases in attenuation are an indicator of non-optimal fermentation, which can be caused by factors such as a non-optimal fermentation temperature, the presence of non-fermentable sugars, a low fermentative capacity of the yeast or the fermentation-inhibiting effects of bioactive substances. Here, an increase in attenuation—a statistically significant decrease in carbon hydrate content ($p < 0.05$)—upon the addition of increasing amounts of Aronia berries and with longer extraction times was observed (Figure 2). The reference contained 48.5 mg/mL of glucose equivalent, which was reduced to 31.8 and 30.0 mg/mL of glucose equivalent for the DCA 65 and DCA 125 samples, respectively, as well as 39.1 and 34.9 mg/mL for DBT 60 and DBT 45, respectively.

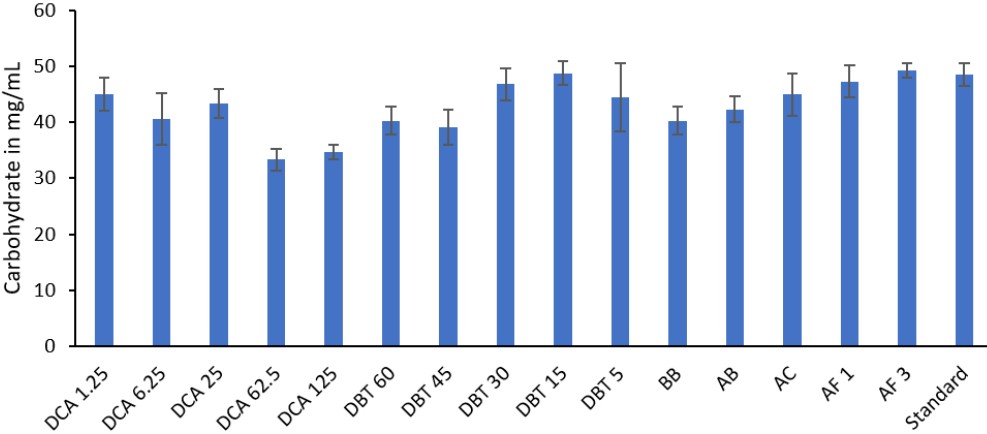

**Figure 2.** Total carbohydrates in Aronia berry-infused beer as glucose equivalents.

Aronia berries could add carbohydrates to the beer and thus affect the apparent attenuation. However, commonly, values of 100 to 200 mg/g of carbohydrates in Aronia berries are reported [15],

which would suggest a very moderate contribution (approx. 1.2 to 2.4 mg/mL for the 12.5 g/L samples) to the overall carbohydrate content in the beer. Here, the result suggests no decrease in the fermentative capacity of the used yeast strain due to the addition of Aronia berries but instead increased attenuation at the higher Aronia berry concentrations of 62.5 g/L and 125 g/L. As the oxygenation levels and pitching rates were kept constant between batches, the acidity of the beer varied only between pH 4.25 and pH 4.81 (see Table 2) and no other variation in the process parameters was applied, the observed significant difference in attenuation seems to stem from compounds extracted from Aronia berries. To the authors' knowledge, no effect of Aronia extract on attenuation has been published; thus, this warrants further investigation.

**Table 2.** pH, Standard Reference Method (SRM) values and EBC color ratings of Aronia berry-infused beers.

| Beer | pH | SRM | EBC |
|------|-----|-----|-----|
| Standard | 4.38 | 6.77 ± 0.01 | 13.32 ± 0.03 |
| DCA 1.25 | 4.4 | 7.90 ± 0.12 | 15.54 ± 0.07 |
| DCA 6.25 | 4.81 | 8.31 ± 0.04 | 16.35 ± 0.24 |
| DCA 25 | 4.42 | 9.01 ± 0.03 | 17.73 ± 0.08 |
| DCA 62.5 | 4.42 | 9.89 ± 0.08 | 19.48 ± 0.07 |
| DCA 125 | 4.34 | 13.36 ± 0.08 | 26.31 ± 0.16 |
| DBT 5 | 4.38 | 8.02 ± 0.09 | 15.79 ± 0.17 |
| DBT 15 | 4.46 | 8.12 ± 0.03 | 15.98 ± 0.06 |
| DBT 30 | 4.35 | 9.69 ± 0.23 | 19.08 ± 0.45 |
| DBT 45 | 4.36 | 11.39 ± 0.16 | 22.43 ± 0.32 |
| DBT 60 | 4.48 | 8.6 ± 0.08 | 16.93 ± 0.15 |
| BB | 4.48 | 8.60 ± 0.08 | 16.93 ± 0.15 |
| AB | 4.25 | 12.03 ± 0.21 | 23.69 ± 0.41 |
| AC | 4.40 | 8.46 ± 0.02 | 16.65 ± 0.04 |
| AF 1 | 4.43 | 7.45 ± 0.11 | 14.67 ± 0.21 |
| AF 3 | 4.39 | 7.03 ± 0.01 | 13.84 ± 0.11 |

*3.2. Color*

The addition of Aronia berries during the boiling stage did increase the SRM and EBC values up to 45 min of boil, after which a decrease in the color rating could be observed. This indicates that the thermal degradation and extraction of the anthocyanins from the berries are competing and that an optimum time for the addition of Aronia berries can be found at around 45 min before the end of the boil step if the coloration of the beer is the desired factor, resulting in an EBC value of 22.43. However, the addition of Aronia berries after the boil resulted in a similar EBC value of 23.69, while addition after cooling only resulted in an EBC value of 16.65, suggesting that higher temperatures significantly benefit the color extraction, even if only used for a short time. The thermal degradation of anthocyanins from black rice was reported to follow first-order kinetics, with a strongly pH-dependent half-life between 2.37 and 0.94 h for pH 4 and pH 5, respectively [16]. This would support the theory that the here-observed approx. 30% lower EBC in the samples boiled for 60 min (BB), compared to that in the samples where Aronia berries were added after the boil (AB), can be explained by the thermal degradation of the anthocyanins. Overall, the color of the beer ranged between EBC 13.84 for the shortest infusion time and EBC 26.31 for the large added Aronia berry amounts. These EBC values are common for wheat beers and pale ales, suggesting neglectable to small changes in the overall color due to Aronia berry infusion.

### 3.3. Total Phenol

Interestingly, the total phenol amount in the sample (Figure 3) was not affected by the boiling duration and showed no direct correlation to the color rating, suggesting that the color and total phenol content cannot be easily and directly corelated and used for total phenol estimation. Likewise, no direct correlation between the antioxidative capacity and total phenol concentration could be established. As higher amounts of added Aronia berries do increase the total phenol content, it can be concluded that the maximal solubility and extraction rate are not limiting factors in the phenol extraction but rather that the extraction is rapid and complete due to the high solubility and easy release from the berries.

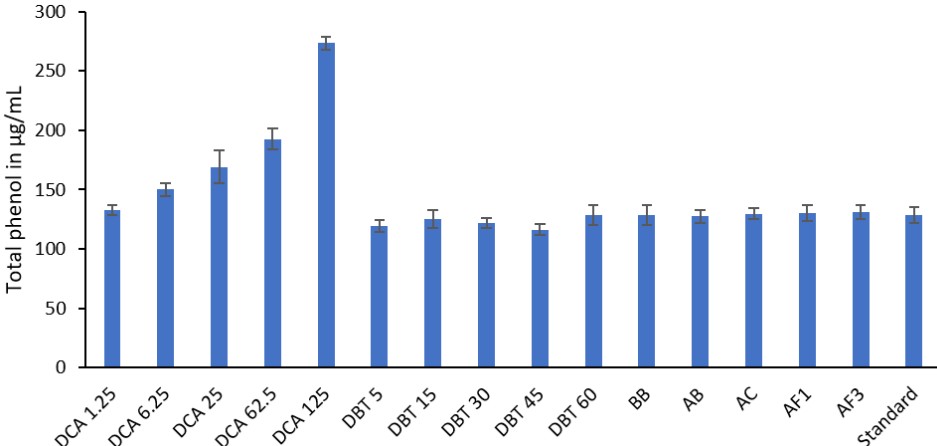

**Figure 3.** Total phenolic compounds in Aronia berry-infused beers.

### 3.4. Ascorbic Acid Equivalent Antioxidant Capacity (AEAC) According to DPPH

Antioxidant capacity was determined via the DPPH assay as ascorbic acid equivalent antioxidative capacity in reference beer without the addition of Aronia berries and all the infused samples. The results are shown in Figure 4, with 2.7 µg/mL in un-infused beer and the values for infused beer ranging from 11.2 µg/mL for the shortest infusion duration up to 84.1 µg/mL for the largest added quantity of added Aronia berries. Larger amounts of added Aronia berries increased the resulting antioxidative capacity, resulting in a strong positive correlation, with a Pearson's correlation coefficient of 0.92 and a $p < 0.009$. The addition of 125 g of Aronia berries per liter showed 84.1 µg/mL AEAC, while the addition of 1.25 g/L resulted in 52.9 µg/mL AEAC. Interestingly, the difference in the antioxidative capacity between the samples where Aronia berries were added before and after the boil (BB and AB) is not significant ($p = 0.48$ with 64.4 µg/mL and 61.9 µg/mL, respectively), while in the samples where Aronia berries were added during the boil, significantly lower ascorbic acid equivalents of 27.8 to 42.2 µg/mL were observed. This suggests that the resulting antioxidant concentration is based on at least two rate constants, likely, one for extraction rate and one for thermal degradation or evaporation of the antioxidant. Overall, the addition of Aronia berries after the boil but before the cooling of the wort proved to be most effective in increasing the beer's antioxidative capacity. The addition of Aronia berries after the cooling of the wort resulted in a approx. 18.6% reduction in the antioxidative capacity. The addition of Aronia berries at different stages of the fermentation, a technique commonly practiced with hops to achieve desired taste profiles, yielded significant lower antioxidative capacity and no discernible difference in astringency according to initial sensory tests and should thus be avoided. Overall, the addition of Aronia berries, even in small quantities, increased the antioxidative capacity from 4.1 fold (after addition on the third day of fermentation) up to 31.0 fold for the addition of 125 g/L of Aronia berries before the boil. The relative contribution of the Aronia berry infusion to the total AEAC would probably not depend on the grain bill or selection of hops. However, different fermentation parameters and lagering times might significantly affect these results, as different fermentation rates

and times could lead to polyphenol oxidation and thus the loss of antioxidative capacity. Additionally, the loss of polyphenols during trub formation as well as adherence to yeast cells and subsequent removal during fining could affect the antioxidative capacity.

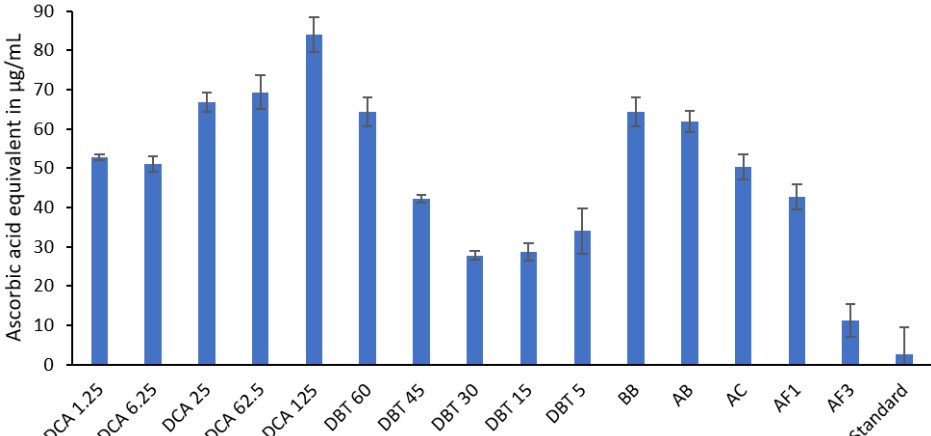

**Figure 4.** Ascorbic acid equivalent antioxidant capacity in Aronia berry-infused beers.

Initial sensory evaluation with a panel of five untrained testers suggested very little to little remaining astringency at the end of the fermentation and no discernable quantitative relationship with the added Aronia berry amount, as well as a logarithmic time dependency of the increase in the astringency with the boil time. The lowest remaining astringency was observed upon the addition of Aronia berries after the boil (AB). Previous studies on the fermentation of astringent fruits have reported significant decreases in astringency after fermentation [17]. In breweries, the flocculation of tannins is a well-established method for reducing astringency. It is possible that the initially dissolved astringent tannins from the Aronia berries contributed to the yeast flocculation and were thus also sedimented. A detailed analysis of the trub (sediment after the fermentation stage) could elucidate the fate of the tannins originating from the added Aronia berries.

Reactive oxygen species (ROS) such as hydroxyl free radicals can initiate radical chain reactions in beer, leading to the formation of aldehydes and ketones, which cause off-flavors and staleness (Kaneda et al. 1989). Additionally, free radical-induced phenol polymerization can lead to the appearance of permanent colloidal hazes from oxidized tannoids [18]. Technical solutions to these challenges rely on forced carbonation and packing under a protective atmosphere to reduce the presence of ROS. However, during the production of beer, the wort has to be oxygenated to facilitate yeast growth, and subsequently, the yeast is only able to reduce the dissolved oxygen to a certain extent before switching to anaerobic fermentation. Therefore, some oxygen tends to remain dissolved in the finished beer and ROS are formed. As shown here, the addition of small quantities of Aronia berries after the boil could significantly increase the oxidative stability of the produced beer by scavenging ROS and lead to an increase in the oxidative product stability. Interestingly, even larger quantities of added Aronia berries do not negatively affect beer color and could be used as a flavoring ingredient in addition to for their stability-enhancing effects. However, one must note that this possibility depends on local laws and regulations, as, for example, the German state Bayern prohibits any addition to beers, while Belgium has a long tradition of fruit-infused beers.

While oxidative stress in humans is a serious health concern, generalized claims of health benefits for antioxidant-rich foods should be only made when substantiated [19]. Thus, here, the possibility of a nutraceutical effect exists, but as no investigation of the bioavailability and resulting reduction of in vivo oxidative stress has been performed, no such conclusion can be drawn.

## 4. Conclusions

The addition of Aronia berries to wort at various stages during beer production can affect antioxidants, color, pH and attenuation. The total phenolic content was only dependent on the amount of added Aronia berries, while the EBC values and antioxidative capacity showed a dependency on the added Aronia amount and time of addition. Briefly, when maximum antioxidant capacity and minimal coloration in the final product are desired, Aronia berries should be added before the boil, while when coloration and antioxidant capacity are desired, Aronia berries should be added to the still-hot bitter wort before cooling, which also resulted in the lowest perception of astringency.

**Author Contributions:** Conceptualization, A.J.; data curation, J.K. and A.J.; methodology, J.K. and A.J.; resources, M.g.C.; supervision, A.J. and M.g.C.; validation, A.J. and K.M.I.B.; visualization, J.K. and A.J.; writing—original draft, J.K. and A.J.; writing—review and editing, A.J., K.M.I.B. and M.g.C. All authors have read and agreed to the published version of the manuscript.

**Funding:** This research was funded by BB21+ 2019 from Busan Metropolitan City.

**Conflicts of Interest:** The authors declare no conflict of interest. This research did not receive any support, financial or otherwise, from industry.

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
