# Peer review of "Antioxidant Content of Aronia Infused Beer"

_fermentation, doi:10.3390/fermentation6030071_

Round 1
Reviewer 1 Report
I don't have any questions about the revised manuscript anymore.
Reviewer 2 Report
I still have concerns about the use of single samples. I wouldn't recommend this for any future studies. There is no way any statistical analysis can be used to partition the analysis and attribute the results to sample when only using one sample.
This manuscript is a resubmission of an earlier submission. The following is a list of the peer review reports and author responses from that submission.
Round 1
Reviewer 1 Report
General comments
The study investigates the effects of Aronia berries on brewing. In particular, the effects of Aronia are evaluated versus sugar content, polyphenol content and antioxidant content. Aronia berries influence the brewing process and change the composition of the final product beyond the simple enrichment in Aronia-derived antioxidants and/or phenols. The study is of interest because could strategically add value to an already existing commercial product. The authors should eventually clarify if they received any kind of support from brewing companies.
The manuscript requires extensive English editing. I suggest the authors to perform this directly with an expert.
Use past tense along all the manuscript
Genus and species must be formatted in italics
An overall conclusion for the work is required. Explain clearly which are the best conditions of fermentation for the production of an aronia beer with a superior antioxidant effect and an acceptable flavour.
Specific comments
Lines 24-26
“Total carbohydrate content at the end of the fermentation indicated no negative effect on the fermentative capacity of the used Safale S-04 dry ale yeast strain, but instead showed increased attenuation for higher concentrations of Aronia”
I would rather suggest:
The increase in Aronia amount increased attenuation showing a positive effect on sugar utilization during fermentation.
Lines 72-73
Avoid “special grade” and “extra pure” and use % where appropriate
Lines 94-96
The flowchart is ok as far as the side panels of Aronia addition are improved and clarfied. This way is rather confusing. I suggest a simplification of the scheme and its association with a table reporting all the operative condition tested with a clear description for each experimental condition. In the table, the same sample identification abbreviations as in the figures must be used.
Table (Example)
|
Sample |
Aronia amount (g) |
Point of addition (according to flowchart) |
Temperature…. |
Parameter 1 |
Parameter 2… |
other |
|
|
DCA 1.25 |
|
|
|
|
|
|
|
|
DCA 6.25 |
|
|
|
|
|
|
|
|
|
|
|
|
|
|
|
|
|
|
|
|
|
|
|
|
|
|
|
|
|
|
|
|
|
|
Line 130
mg/ml, do you mean mg/g?
Lines 128-134
Avoid discussing increase in carbohydrates since it is not observed in the experiments
Lines 132-134
This could be proved by estimating the ethanol content in the different samples. I suggest to insert ethanol data if available
Lines 142-143
In fermentation, the pH change between 4.25 and 4.81 is relevant. Concerning residual sugars and AEAC, DCA 6.25 sample shows indeed a behaviour not in line with the rest of the samples. This aspect should be better investigated or discussed
Lines 147-148
I interpret DAA in figure 2 as being DCA of figure 1. DPT is not reported in Figure one and is reported as DBT in line 191. Please organize a clear table for better interpretation of the experimental scheme and correct figures and graphs accordingly.
As far as evidenced by Figure 2, Aronia has effect on attenuation being the best results obtained in conditions DAA 62.5 and DAA 125
Lines 178-181
If I understood correctly, in figure 3, sample DCA 1.25 (1.25 grams of aronia) has the same phenolic content as the counterparts DBT 60 and DPT BB (both with 10 grams of aronia). This is in contrast with your hypothesis. Note also that the total phenol level is the same also in Standard sample. Do you have any explanation?
Author Response
The authors should eventually clarify if they received any kind of support from brewing companies.
We thank the reviewer for the suggestion and have added one sentence to the conflict of interest statement, declaring that no support from industry was received.
The manuscript requires extensive English editing. I suggest the authors to perform this directly with an expert.
Use past tense along all the manuscript
Genus and species must be formatted in italics
We apologies for the oversight, have fixed the instance and changed all mentions of ‘Aronia’ to Aronia berries for clarity and have submitted the manuscript to language editing.
An overall conclusion for the work is required. Explain clearly which are the best conditions of fermentation for the production of an aronia beer with a superior antioxidant effect and an acceptable flavour.
We have added a conclusion at the end of the manuscript according to the reviewers suggestion. We also have added a few more details of the description of the initial sensory evaluation.
Lines 24-26
“Total carbohydrate content at the end of the fermentation indicated no negative effect on the fermentative capacity of the used Safale S-04 dry ale yeast strain, but instead showed increased attenuation for higher concentrations of Aronia”
I would rather suggest:
The increase in Aronia amount increased attenuation showing a positive effect on sugar utilization during fermentation.
We thank the reviewer for her/his comment and have amended the manuscript accordingly
Avoid “special grade” and “extra pure” and use % where appropriate
We have amended the manuscript accordingly and removed all word-based purity descriptions.
The flowchart is ok as far as the side panels of Aronia addition are improved and clarfied. This way is rather confusing. I suggest a simplification of the scheme and its association with a table reporting all the operative condition tested with a clear description for each experimental condition. In the table, the same sample identification abbreviations as in the figures must be used.
We thank the reviewer for the comment and have simplified the flowchart and added a table for clarification.
mg/ml, do you mean mg/g?
We apologies for the oversight. Some works cite the carbohydrate concentration in pressed juice from Aronia berries in mg/ml, but this particular reference is in mg/g carbohydrates from fresh fruit. We have amended the manuscript accordingly
This could be proved by estimating the ethanol content in the different samples. I suggest to insert ethanol data if available.
Unfortunately, we have only relative imprecise ethanol estimates derived from the original gravity and sugar analysis. We apologize for the oversight and ask for the reviewer’s kind understanding in this matter.
In fermentation, the pH change between 4.25 and 4.81 is relevant. Concerning residual sugars and AEAC, DCA 6.25 sample shows indeed a behaviour not in line with the rest of the samples. This aspect should be better investigated or discussed
We thank the reviewer for the suggestion. AEAC is linearly correlated with Aronia berry concentration with a Pearson coefficient of 0.92 and a p of 0.0088 (the information has been added to respective section). For residual sugar, the sample DCA 6.25 is not statistically significantly different than DCA 1.25 or BB (effectively DCA 12.5) with p values of 0.3 and 0.9 respectively. We therefore don’t believe that the data would support the statement that DCA 6.25 is statistically significantly different to warrant a separate discussion. Similarly, we agree that a difference of more than 0.5 pH can be very relevant, we unfortunately have not enough data to confidently discuss these differences based on statistically significant figures. We would ask for the reviewers kind understanding in this matter.
[A private thought, separate from the official replies to the reviewers comments: Couldn’t the small difference in DCA 6.25 be the result of not perfectly closed containers? If CO2 escapes, the pH would be slightly higher, oxygen could ingresses, AEAC could be slightly lower and sugar utilization could increase.]
I interpret DAA in figure 2 as being DCA of figure 1. DPT is not reported in Figure one and is reported as DBT in line 191. Please organize a clear table for better interpretation of the experimental scheme and correct figures and graphs accordingly.
We thank the reviewer for the careful reading of the manuscript and sincerely apologize for the oversight. We have amended the manuscript accordingly.
As far as evidenced by Figure 2, Aronia has effect on attenuation being the best results obtained in conditions DAA 62.5 and DAA 125
We thank the reviewer for the comment and have rewritten the relevant section
If I understood correctly, in figure 3, sample DCA 1.25 (1.25 grams of aronia) has the same phenolic content as the counterparts DBT 60 and DPT BB (both with 10 grams of aronia). This is in contrast with your hypothesis. Note also that the total phenol level is the same also in Standard sample. Do you have any explanation?
Frankly: not. The analytical method was properly validated, all measurements were withing the linear range and quantitative range of the assay. Analyzed on the same day, no time difference between different samples, same reagents, same instrument. The measurement was even repeated 9 times (!) and we saw relative standard deviations between 2 and 8%. And like you have pointed out the non-significant difference between the standard and samples with 12.5g/l Aronia is surprising. The BB sample can be considered DCA 12.5, even with this sample as difficult to explain outlier the Aronia concentration total phenol correlation is very clear with a Pearson correlation coefficient of 0.979 and a p of 0.00062. Thus, based on the data we can only confidently discuss correlation, but nothing else.
Reviewer 2 Report
The manuscript describes an experiment where a berry is added to enhance the antioxidant properties of beer. While the authors demonstrated the contribution of the beer added so additional antioxidant properties, the experiment uses a single malt sample, in a single type of brewing style. I would like the researchers to go back and demonstrate the same results with many different type malts, different hops in different beer styles using different samples of the berry. This would prove their experiment is valid. They can use a smaller mash size rather than a larger mash used here. The small lab scale mash would allow more samples to be tested simultaneously. Thereby identifying the true contribution of the berry when we know different malt and hops also contribute varying level of antioxidant properties.
Author Response
We thank the reviewer for the reading of the manuscript and her/his suggestion. The chosen batch size of 800ml per boil was found to be acceptable in terms of material consumption but especially for variation due to volume loss to evaporation during the boil. Furthermore, were smaller batch sizes and thus smaller Aronia berry amounts requiring small number of individual berries, thus posing the risk of larger variation in the phenol and antioxidant results. The stated purpose of the research was to identify suitable process conditions for the infusion of beer with Aronia berries, analyze contributions of the Aronia infusion to antioxidative capacity, total phenols and do an early assessment on possible negative resulting flavor. Investigating different types of malts would not serve this purpose, nor would different hops or mashing styles, as all these could affect the antioxidant and phenol content in the resulting beer originating from the malts and hops, not from Aronia berries. There are already interesting papers discussing antioxidants and phenols in various malts and mashing styles. We would particularly recommend Koren et al, “Study of antioxidant activity during the malting and brewing process” 2019 (https://doi.org/10.1007/s13197-019-03851-1). Similarly, the investigation of the antioxidant capacity of the berries of various Aronia cultivars is in itself interesting but not related to determining suitable process parameters for the beer infusion. Here too existing work could be of interest: Wangesteen et al. ‘Anthocyanins, proanthocyanidins and total phenolics in four cultivars of aronia: Antioxidant and enzyme inhibitory effects’ 2014 https://doi.org/10.1016/j.jff.2014.02.006. We therefore agree in principle that the reviewers suggestions are on their own worthwhile avenues of research, but believe that they lie beyond the scope of this research. We ask for the reviewers kind understanding in this matter.
Reviewer 3 Report
The article entitled "Antioxidant Content of Aronia Infused Beer", is well written and organized. However, the measurement of antioxidant activity by a single method is not sufficient. The authors need the manuscript better by adding more antioxidant methods, namely FRAP or ABTS.
Author Response
We thank the reviewer for her/his time and suggestion. We agree that accurate estimation of actual antioxidative potential in vivo is very challenging and benefits from in vitro measurements with multiple assays. The same holds true for comparison of various extraction methods where occasionally significantly varying antioxidative capacity is reported.
We have chosen to use a single assay for the assessment of antioxidative capacity in Aronia infused beer because it has been demonstrated that DPPH, FRAP and ABTS assay results for Aronia berries are strongly corelated. Hwang et al have shown correlation coefficients of 0.9241 (DPPH-ABTS) and 0.9605 (DPPH-FRAP) respectively (Hwang et al. ‘Radical-scavenging-linked antioxidant activities of extracts from black chokeberry and blueberry cultivated in Korea’ 2014). For a large variety of commercially available beer it also has been shown by Zhao et al. that DPPH - ABTS and DPPH – FE3+ reducing power are also highly corelated – with a correlation coefficient of 0.973 and 0.950 respectively (Zhao et al. ‘Phenolic profiles and antioxidant activities of commercial beers’ 2010). Thus, we do not believe that additional assays will contribute significantly to stated purpose of this project, which is establishing optimized process parameters for Aronia infusion of beer and assessing which condition offers largest gain in antioxidative capacity. However, we agree with the reviewer that if claims for health benefits due to increased in vivo antioxidative capacity would be made, then a more in-depth antioxidant analysis would be needed. We would ask the reviewer for his/her kind understanding the matter.
Reviewer 4 Report
- The full name of DCA should be noted under Figure 1.
- The full names of DAA, DBT should be noted under Figure 2 and table 1.
- What is the statistical design for the current study?
- Where is the correlation analysis in the result? As you mentioned “positive correlation” in the abstract. How much is the correlationship?
- What’s the deviation for the data in table 1?
Author Response
1.The full name of DCA should be noted under Figure 1.
2.The full names of DAA, DBT should be noted under Figure 2 and table 1.
We thank the reviewer for the careful reading of the manuscript and pointing out the naming inconsistency. To address this we have changed the overall layout and included a table with abbreviations, their full names and process conditions and also have unified the abbreviations in the figures and tables.
3. What is the statistical design for the current study?
We apologize for the oversight and have added a small description under 2.6.
4.Where is the correlation analysis in the result? As you mentioned “positive correlation” in the abstract. How much is the correlationship?
The mention in the abstract was intended as a brief summary of the in depth discussion in the results. We apologies for using the term without showing the actual value in the results. We observed a positive correlation of a pearsons r=0.92 with a p<0.009. We have added the information to the discussion section and apologize for the oversight.
5.What’s the deviation for the data in table 1?
We thank the reviewer for the suggestion and have added the standard deviations to the table and added clarifications to the method section.
Round 2
Reviewer 2 Report
I appreciate the authors response to my comments. I feel the paper is still very limiting with only one type of brewing process used. Yes the addition of these berries did increase the antioxidant properties of that laboratory produced beer. Ask yourself, would the results would be the same with different brewing styles, different types of malt, different types of hops? Different fermentation processes would have different DO levels, how would this impact on the effect of the antioxidant levels. How would the antioxidant levels change in storage in can, bottles or kegs? This needs to be mentioned in the Abstract, Discussion and Conclusion, that the results are from just a single brewing process, with specific malts and hops, and the results could change considerably in different brewing processes.
Author Response
We thank the reviewer for his/her time and valuable comments. The antioxidants in beers with different malt or hop types and mashing styles could be indeed different. However, the contribution of the Aronia berries would be likely same, as to our knowledge there has not been any reports for example on an observed effect of hop type on the extraction of antioxidants from fruits. Also, to our knowledge no mechanism exits, where any differences found in the various malts could affect the extraction rate of antioxidants from a berry. From our observations the largest factor would be boil time which can vary widely even in the same beer type as it controls a variety of factors: original gravity of the wort, DMS reduction, alpha acid isomerization and Thiobarbituric acid index, among others. Therefore we do believe that the investigation of boil time – antioxidative content relationship performed within this work gives a good initial assessment of effects likely observed in a wide variety of processes with different boiling times.
It also should be noted that to our knowledge no research has been published describing any inherent effect of type of storage container on the antioxidant capacity of beer. Instead a comparison between PET and metal Kegs by the company ‘Sauer & Hartwig Technologie’ showed no significant differences in the sensory evaluation of the differently stored beers.
The change of antioxidative capacity during fermentation and storage is a very wide and well researched topic, intrinsically related to the ageing and shelf life of beer. As the reviewer suggested this is an interesting topic. However, as the focus of this work was on the establishment and assessment of an infusion process for Aronia beer, we feel that this would be beyond the scope of the presented work and more suited for future research. Especially when suitable process conditions during the mash and boil should be identified, an investigation of fermentation processes would not help in identifying these process parameters. Here initial oxygenation, pitching rate, and initial yeast propagation speed would affect the DO in the bitter wort and thus potentially the antioxidative capacity in the final product. However, we respect the reviewers opinion and realize interested readers might be also interested in the points raised by the reviewer, therefore a short discussion and clarification was added to the abstract (Line 19), discussion (Line 213) and conclusion (258).